# The limits of learning engagement and academic leadership within the higher education digitalization process - analysis by using PLS SEM

**Ioana Gutu** [ID][☽]*, **Camelia Nicoleta Medeleanu**[☽], **Romeo Asiminei**[☽]

Department of Sociology and Social Work, "Alexandru Ioan Cuza" University of Iasi, Iasi, Romania

[☽] These authors contributed equally to this work.

* gutu.ioana@yahoo.com

**Data Availability Statement:** All relevant data are available on OSF: https://osf.io/ja8ux/

**Funding:** The author(s) received no specific funding for this work.

## Abstract

There is convincing evidence that the learning environments digitalization of tools and equipment ultimately results in the speed and depth learning involvement of academia members, by raising attainment of each of the digital learning experiences. The majority of the research that was conducted on the topic of enhancing the digital skills of learners, which would ultimately lead to an increase in their active engagement, was conducted on students in primary and secondary education, leaving members of higher education outside of the scope of the study. Given the uninterrupted search for academic performance and innovation, the current research considers the technological changes that lead to the transformation of the traditional academic learning environments as previously known. The current paper considers the changes in the learners' engagement in the context of the dually digital transformation of the higher academic multi-institutional digitally-learning enhancements. An important factor to be considered regards the leadership evolution (in terms of teaching) that over time, led to a different speed contextual shift, according to its effectiveness, leading to higher or lower students learning (dis)engagement. The current manuscript aims to examine how the higher education digitalization levels could affect the student's learning engagement, under the close monitoring of the academia leadership styles practice. Data collection and analysis implied at first a qualitative approach by issuing an online-distributed survey that resulted in a number of 2272 valid responses. After performing structural equation modelling and proving a valid assessment tool, the analysis resulted into statistically proving the validity of two main hypotheses according to which students learning engagement has a positive effect on the practice of academic leadership. Additionally, results emphasized the fact that higher education digitalization has altogether a negative effect of students learning engagement. Consequently, the current study stresses on the importance of different peers' categories in the context of higher education institutions performance, with an emphasis on the different levels of students' engagement and the leadership styles evolution and practice, aspects uniformly developing within a continuously digitally transformation of the higher education environment.

**Competing interests:** The authors have declared that no competing interests exist.

## Introduction

Once the rate of digital change inside the higher education institutions becomes slower compared to the rate of digital evolution outside, the failure of the academic teaching process is considered to be near [1]. Given the speed of evolutionary digital transformations and all related technologies that all the learning environments are subject to, the current business environment dynamics has become problematic. Forthcoming, all the organizations, and especially the public sector representatives, are highly considering arising opportunities in favor of adopting and implementing digitally transforming technologies that would result into increased organizational agility and flexibility. Labor market demands make higher education institutions active seeking subjects of digital adoption and change, especially since public sector institutions are known for lack of administrative flexibility, resources, data availability and management capabilities [2, 3]. Despite the given setbacks, implementing digitalization within the public higher education institutions holds numerous benefits, leading to increased transparency, innovation and environmental support, staff and peer engagement and participation invigoration [4, 5]. As a result, the higher exposure of labor and industries sectors to digitally transformed work environments have led to a high demand for higher education institutions to quickly adapt to digitalization endeavors and provide education environments that would increase the students will and engagement to study, thus taking advantage of the leadership traditional and newly adapted stimuli.

Not until recently, the once disempowered power of leadership command became acquired dominant power across the entire business and higher education worlds, acquiring powers for transforming work landscapes and becoming a buzzword within the research topics specific to social sciences. Despite its origin as a converging variable used in solving action and conflict, singular and collective issues, within the global digital landscape, leadership (and it's half twin, followership) have reached to define the activities nature and traditions of each and every institution umbrella it has penetrated. The accent regularly falls on the transformational and transactional leadership practice, since under a stable working environment, each of the two styles has the ability to predict performance and enhance change. Within the higher education institutions, the importance of different leadership styles practice falls on the personnel (seen as educational staff) endeavors, wick are likely for the large majority of cases to be used for increasing the active students learning engagement. Within the given conditions, the symbiotic relation involving teachers and students falls under the effects of the digitalization instruments specific to each context, the implementation of teaching and learning instruments on one side, and the adoption and use of such technology on another side, falls under a major pressure form the entire peer society.

Work engagement and results has been subject to managerial and social studies for decades, but not until recently, it was considered to be a valuable asset the information in regard to the practice of leadership in the context of digital solutions, along with data resulting from students involvement and engagement, following a leading research teaching methodology perspective [6, 7]. The nature of such an angle requires a new didactic design based on the importance-performance of all the given variables, with an accent on a content analysis on the students' entry conditions and indicators; preliminary results suggest a binding need for the higher education academic environment to be subject to a massive redesign of specific teaching and learning angles.

Previously analyzed as a participation process within the higher educational processes, learning engagement has been mainly characterizing indoor practices, leading towards measurable higher educational quality results [8]. The current approach acknowledges learning engagement as students' efforts resulting into higher and/or lower quality results, achieved

through devotion towards the higher educational purposes, directly contributing to the targeted outcomes.

Aside from teaching staff status quo past and present performance requirements, from the students perspective, a higher education environmental change would require data revealing further performance knowledge and abilities, based on business and institutional active needs, reality-anchored; such a process would require an adequately adoption of digital solutions for both academic institutions and its members (teaching and learning), allowing an optime engagement within the fully schedules of the higher education processes [9]. In short, given the premises, the current research addresses the following research question: *which are the contextual premises for the higher education digitalization and academic staff leadership in order to support and grow high levels of students' engagement*?

The vast majority of business and educational literature in regard to technological implementation and change focuses of key aspects such as advantages and disadvantages for the organizational framework itself, failing to address the working and peer staff perspective. For this reason, efforts must be directed towards more analytical results that would explain the relationship involving the digitalization effects on employees-to-be, therefore higher institutions digitalization effects on leadership practice and its effects and students' engagement and its evolution, are deemed as necessary.

To answer the previous question, a thorough discussion on the evolution of digital advancements, from digital transformation to digitalization needs to be addressed. Explicitly, it will enable the importance of digital framework evolution for the higher educational institutional record. Further, academia learning engagement must be assessed, given the high-speed technological advancements, considering all of its components. Higher education digitalization is considered to be a motivator for its students, but efforts must be exerted towards explaining the dynamics of teaching leadership and followers' engagement.

The current research considers the self-assessment of actively academia enrolled students, who provided inputs in regard to teaching staff leadership and the digitalization tools available and used within their undergoing activities, leading towards a degree of individual learning engagement specific to the institutional background. The empirical results lead the authors towards a connection to previous theory, focusing attention of the importance of adapting digitalization to teaching and leadership processes, ultimately resulting into a higher degree of actively enrolled students learning engagement, with an active consideration on transformational and transactional leadership styles.

The current research points up at first the evolutionary perspective of learning engagement within the higher education institutions, emphasizing the role and effects of institutional digitalization efforts. Further, different leadership practiced styles and individual learning engagement of students' perspective has been discussed under a compounded literature network assessment. Methodology aspects reveal the complexity of the database by using the Smart PLS Software (v. 4.0.0), testing the general and in-depth hypotheses and revealing contextual valid background for the initial assumptions of the current research, and providing data support for the previously exposed research question. After a complete analysis and discussion of the results, there were considered the practical and theoretical implications of the resumed data, followed by limitations and research final remarks.

## Learning engagement and higher education digitalization

The increasing use of digital technologies across the entire framework specific to the higher education environment is steadily increasing the digitalization into taking the form of a pillar of the uninterrupted search for academic innovation and performance [10, 11]. The last

decade has witnessed a change in the higher academic business model, driving organizations to perform a symbiotic process implying the strategies redefinition and the actively enrolled academic engagement [12–15]. The newly created organizational context is highly subjective to the practice and creation of leadership styles strategies that play a critical role in engaging digitalization and the students learning engagement characterizing the respective academic environment.

By following the pattern of logic, based on previous research, a distinction must be made between three terms that are often mistakenly referred as with the same meaning, given the fact that they are built on another; therefore, in between digitization and digital transformation, resides the word digitalization [16], as referring to the aid provided by the digitally available information as for improving previous automated endeavors, as for the current case, teaching and learning. By using the previous data, it would result the fact that higher education digitalization refers to the process where traditionally hand-written documents (e.g. tests, courses, homework) are scanned and made as online available, thus favoring of new workflows that would improve the time and quality of higher education actions and results.

Due to the wide variety of disciplines that currently use digitalization as a transformation process, the three terms previously mentioned are used without coherence and consistence, without having a common understanding previously established.

Learning engagement has been previously analyzed under the identity of a process of participation in effective higher educational processes and practices, mainly indoors, leading towards measurable high quality learning results [17, 18]; for the purpose of the current study, learning engagement is rather defined as the students' efforts and results quality, achieved through devotion to the higher educational purposes, thus directly contributing to the targeted outcomes.

According to literature [19], *learning engagement* can also be understood as a multidimensional construct with behavioral, emotional, cognitive and social components [20] but also as a higher education institutional process of empowering students to achieve the desired outcomes, a process designed to deliberately leave a high impression on their learning experience. A new vision over learning engagement was achieved [21] by combining previous views from literature; learning engagement is described as a higher educational complex process characterized by indicators as students time, effort and devotion to activities that would ultimately fulfill the preset/desired performance indicators that are coordinated and calculated not only on a personal, but also on an institutional level.

Through digitalization, the students learning engagement acts as a booster for higher education institutions to gain competitive advantage and recognition. By being actively engaged, students will be highly motivated on both–educational and personal levels; if managed properly, engaged students start acting as unique factors that from the competitor Universities, become very hard to duplicate and imitate, thus they should be treated as important assets for every institution the process involves [22]. Nonetheless, engaged students provide higher education institutional background with valuable information in regard to organizational commitment [23, 24].

Across global studies, there is empirical evidence that screen time is not a direct indicator of higher education students' performance [25], meaning that longer hours of television and gaming are associated with lower rates of academic performance and engagement for higher education students. It is important to note that the digital engagement does not have an exclusively linear effect on academic engagement and performance [26] under the same general idea, a study published in 2017 [27, 28] reported that screen-based activities were highly associated with increased higher education engagement and performance, with a maximum of 4 hours/day.

Academic engagement within higher education institutions can also be observed under the form of digital engagement, considered to be a valuable resource (given the studies levels and concepts complexity targeted and nurtured within higher education system) if and when it comes under another form, different to social media and associated activities, leading towards small positive results when engaging and measuring academic achievements [29, 30]. According to literature [31–34] suggests, social networking is weakly correlated to academic performance, while social media usage with academic purposes within a formal environment is positively associated to studies proportionality, thus leading towards a higher academic engagement. Studies have shown that by gaming and social media, some of the students gained high-level digital skills, but the pedagogical assessment and challenge for assimilating such qualities still resides across the entire higher education system [35, 36]. Moreover, the use of social media groups was a significant predictor of grades [37], while reviewing lecture slides/recordings, reading supplementary content, and using course blogs/discussion boards were not. As for the cognitive ergonomics, it has been proven that higher education digitalization manifested through multiple distractions and technology forms and channels are a negative influence for learning engagements, therefore educational demands must be carefully assessed and further' used [38, 39]. This affirmation is being followed by another research results where multitaskers (doing simultaneous activities) inter-related to digital academic endeavors, leads to a lower engagement (manifested as study-related behavior) and academic performance [40–43].

Another facet of digitalization specific and not only resuming to higher education systems is (academic) networking, a variable based on college student samples that was proven to be negatively related to academic engagement; it is deemed as necessary to mention the fact that results in regard to learning engagement in the context of higher education environments vary in accordance to the objectivity of the measures used [44] the higher the objectivity being leading to a less evident relation involving the negative effects over time that are mentioned above [45]. Social networking, when leading to fatigue, will conduct to an accelerated lowering of academic engagement [46], thus leading to sleep disorders and energy depleting processes [47]. For this reason, especially after the 2020 pandemic, the higher education academic environment must be held within an equilibrium in regard to digitally liable enhancements, especially when academic performance is decomposed within learning engagement and other contextual learning factors.

Another important factors in regard to learning/study engagement are the individual features that results into the uniqueness of every student; here we take into account features such as identity or personality, additional to a series of individual pre-existing dispositions towards supplementary work, attention to details or family-related cognitive resources availability. Such situations enhance the perceived learning engagement achieved within higher education institutional processes, leading towards accomplishing higher quality academic performance and related outcomes [48–50]. As per additional individual features, one would note the extroversion scores that are described across literature to benefit from the higher education digital media usage in terms of students engagement, while on the opposite pole, students with non-adaptive skills prefer remote and/or online activities and academic interaction, resulting into a lower engagement and a poorer performance [26, 51, 52].

Learning engagement as considered for the current research, consisting of three main variables–as autonomy, social support and educational activities engagement, is being measured with the usage of four domains–behavioral, emotional, cognitive and social [53]. A sound learning engaged behavior is translated through students individual efforts to accomplish tasks with perseverance, complete homework and put efforts into course/class participation, therefore emphasizing collaboration, social interaction, skills development, performance achievement [54, 55]. Regarding learning motivation, literature [56] showed that students' intrinsic

motivations predict engagement, while extrinsic motivations predict use. As for emotional engagement, it relates with assessing students' responses in regard to learning activities, measured in terms of interest, happiness, anger, anxiety in regard to usual of challenging specific activities provided by the higher educational environments. Also, it was highlighted [56] that students are more creative when there are in positive emotional engagement. Cognitive higher education students' engagement explains one main variable–the students cognitive interaction measured in terms of willingness and strategies towards achieving their academic goals. The measure of the effort of an individual in the process/ as for interacting with the other is commonly referred to social interaction, the higher the social engagement, the higher the academic results, regardless the difficulty of the occurring processes.

Learning engagement levels have been identified within literature as highly related to higher education students cognitive and practical achievements [57]. To this end, the entire system digitalization not only resides within administrative or clerical duties, but actively involves academic endeavors too; for this reason, across literature was revealed that digitalization practices (such as digital environments usage, digital teaching and learning platforms), when scheduled and time researched properly, lead towards increased levels of students engagement, measured through skills acquisition, learning achievements and academic writing skills [58–60]. Moreover, digital teaching and communication platforms (such as Blackboard, Moodle, Easy Class) proved increased performance in bachelor students both behavioral and cognitive engagement levels, skills and performance, as compared to students that only used a regular (non-digital) study environment [46, 61]. Given the previous arguments, the following hypotheses arise:

H1. Higher education digitalization has a negative effect on learning engagement

## Learning engagement and leadership

Given the fact that leadership is one of the most studied subjects across world literature in regard to organizational behavior, a thorough review of the leadership theories is not considered to be deemed as necessary but is worth to mention that transformational leadership appears to be the most pervasive subject among all [62, 63]. Whatever the leadership theory assumed to be bet fit for an organization, there is an overarching assumption that the relationship developed between a leader and a follower has the ability to predict outcomes at different levels- individual or organizational, an idea that can be explained through the fact that leadership is an antecedent of a large array of factors, out of which individual engagement prevail [64, 65].

The concept of engagement was for the first time discussed within the literature around the 1990's and was expressed as the way people interact and convey on a physical, cognitive and emotional levels, in regard to the achieved performances [66, 67]. Once with the terminology, the business and educational environments, along with human resource management also evolved, thus favoring the creation of an interchangeable meaning for organizational engagement such as referring to personal, role, work, employee or job. Literature also enhances a psychological view over organizational engagement as referring to the readiness for experiencing and engaging distractions while holding the role of a societal system. Further assessments [68, 69] define organizational engagement as a positive state of mind, developed in regard to the individual and/or team collaborative relations, characterized by a sense of significance, full concentration and personal absorption while working (study); the process would end with a rapid passing of time, resulting into a difficult detachment from (home)work. From an evolutionary perspective, engagement views migrated from personal to an organizational perspective, resulting into a cognitive, emotional and behavioral state of mind related to the organizational resources and results [70].

The modern perspective on learning engagement may be the key for understanding the role and implications of leadership over the students states of mind, behaviors and related outcomes, thus implying different teaching roles and perspectives.

Transformational leadership and learning engagement are a relation that often catches the reader's eye, especially when recurring to the higher education world, by potentiating a sense of security among the studying masses. Literature has shown [71, 72] that the transformational leadership proved to have a sense of emotional support in the eyes of their followers, while their efforts recognition lead towards feelings of secure attachments within the higher education organizational environment, followed by psychological shifts of focus involving gains instead of losses; studies assimilate vision and vigor, but only within a scenery of fast and fair promotion and public recognition of their merits; moreover, supportive transformational leadership is not presented as directly related when about students anxious behaviors, while none of the transformational leadership forms and related conduct was proven to be related to the individual students' self-esteem [73, 74].

Another research [75] studied whether transformational leadership is related to any level to learning engagement, and whether the students' engagement–higher educational leadership relationships is stronger when the students (followers) personal characteristics are higher and/or lower. Results have shown that the students-rated transformational leadership is strongly correlated with the students' engagement (in regard to their efforts to learn) on one side, and teaching practices of transformational leadership led towards is highly correlated with the students self-rated personal characteristics. Moreover, higher education students self-rated characteristics were proven to have a positive effect on their learning engagement, leading towards a clear prediction of the higher-education teaching–students engagement relationship slope. As literature has previously shown, within higher education institutions there is a symbiotic and influential relationship that involves teaching staff as leadership representatives on one side, and students learning engagement levels and engagement behaviors on a subsequent level.

Studies in regard to transformational leadership and learning engagement were performed in regard to activities performed at a day-level, in regard to self-efficacy and optimism [76]. Results have shown that there is a positive relationship between transformational leadership and students learning engagement on a day-level basis; moreover, self-efficacy and optimism also correlated highly with engagement. However, a general basis (different from daily) there was not a direct connection (non-significant) between transformational leadership and learning engagement [77, 78]. A nomological perspective over the conceptualization relationships including transformational leadership and learning engagement (thus referring to measures such as organizational support, social support, autonomy, engagement and satisfaction); results show that higher education transformational leadership in regard to social support were mediated by the student's engagement to study [79]. On another perspective [80, 81] learning engagement plays the role of a mediator between transformational leadership and students extra-role performance. Additionally, it was proven that responsibility, meaningfulness and innovative behavior that explain learning engagement in this particular research, are characterized by specific indirect effects in regard to transformational leadership [82]; moreover, the paths from transformational leadership to learning engagement were also characterized by positive and meaningful effects. Moreover, it was argued that leadership plays an independent role when predicting knowledge creation [67, 83], but it is significantly smaller compared to when assessing the indirect impact in regard to the learning engagement variable. On another view, the indirect effect of transformational leadership was assessed in regard to success, the model being considered a better fit when mediated by learning engagement [84, 85]. On another view, transformational leadership was indicated to be significantly associated with an

increased students learning engagement, leading to an additional development of organizational relationship [86, 87].

When assessing conceptual studies in regard to leadership and engagement, literature has shown numerous results that present academic leadership to play an important role when creating a teaching environment where students feel energized and actively engaged within the student-teacher academic activities. There were identified four recommendations for increasing the trust and enhancing this relation; it involves the use of personal resources, the organizational design for meaningful and motivational work, coaching and support activities coming from the leadership figures, along with active rewarding for the efforts provided by the students [88].

If keeping this train of thought, it was emphasized that by providing appropriate support and just resources, the higher education institutions organizational environment could be the right setting and contributor for the students' energy and dedication, thus favoring the creation of a persistent and vigorous environment that would further nurture the students learning engagement [79, 89, 90]. An engaging climate produced by the academic community should be able to develop features as self-awareness [91–93] meaning a deep understanding of the depth of their leadership actions and information provided, therefore becoming a conceptual cornerstone for the emotional intelligence of the most profoundly learning engaged students; leadership has the ability to promote engagement among the actively enrolled students, by providing attention to the details of their basic needs; by showing a willingness to respond to these issues, the higher education organizational teaching and learning environments will not become liable for the lack of engagement of students which have not inwardly focused on their personal development, leading to lack of task completion or lack of performance achievements.

As previously proven, there is a consistent portion of literature that regards the relationship between teaching leadership and students learning engagement, leading towards the belief that this is a traditional, routine assessment for most of the educational organizational environments. Moreover, there is little evidence that surrounds literature in regard to the leadership styles, such as transactional or laissez-faire [94, 95] in the context of academic leadership-learning engagement relationship, under the motivation that followers respond differently to related leadership characteristics in association to academic learning engagement across time. In addition, literature illustrates a lack of abundance of recent approaches in regard to the leadership-engagement relationship in the context of higher education organizational environment. Some of the approaches that examine the daily view of the leadership -learning engagement relation proves the higher educational teaching environment to be beneficial, while on a large scale, different organizations call for the adoption of different leadership styles, in accordance to the peculiarity of each situation. Therefore, a gap within the literature has been identified–namely, based on the necessity of temporary leadership approaches within the higher education systems, the use of particular leadership styles–apart or combined–and their effects in regard to students learning engagement, has not been a subject for study. with one exception [65], none of the authors considered for research the subject of multiple or interchangeable leadership styles to affect the peculiarities (as autonomy, social support or students' engagement) or the general outcomes of students learning engagement. The study in regard to the usage of a dyad of leadership styles could provide the literature with a better and richer understanding of the complex relationships that imply teaching–leadership and learning engagement, with an accent on the implications on the followers' attitudes and academic performance.

Most of the studies agree on the fact that within a higher education teaching and learning environment, leadership is mostly correlated or plays the role of a mediator in regard to the

students learning engagement [96]. Another gap still resides in this regard: there is little to nonexistent research as to whether the positive relationship can be observed as functioning across time; moreover, the question of which of the two variables–academic leadership and learning engagement causes the other [97, 98]. Across literature, the large majority of the research findings in regard to (teaching) leadership and (students learning) engagement can be mostly described as narrowly focused, thus presenting the depths of this relationship as inconclusive.

As previously mentioned, many other leadership styles–apart for Bass and Avolio's transformational leadership, have been a subject for study in connection to the psychological mechanism underlying the higher education students learning engagement depths and effectiveness [96, 99]. Certain approaches emphasize the role of transactional leadership in engaging actively enrolled students by clarifying their roles and providing active recognition of their efforts, by implementing the view within the core of the higher education organizational culture; additionally, it was noted that the transactional leadership effectiveness in regard to the students (as followers) learning engagement could suffer variations in accordance to other organizational and personal factors. If empowerment is a feature related to the transformational leadership that positively affects learning engagement, the identification with the leader (which plays a transactional role), along with psychological impersonation is highly specific to the transactional leadership roles actively played within the higher education institutions [100–103]. Moreover, learning engagement has been often associated with the task-oriented transactional leaders, leading towards an increase in the autonomy and social support, especially when referring to humanistic study disciplines within higher education institutions [104].

As for laissez-faire leadership practiced within higher education institutions, the large majority of the studies provide an inverse association with learning engagement, since this particular leadership style is mainly counterproductive, presenting an extended lack of personal initiative, leading towards negative results in regard to the academic performances of the enrolled students.

It is notable that the organizational dynamics is heavily influenced by the leadership style practiced by tutors and academic staff; there is a myriad of situations that could result into counter-productive actions and activities that would lead to a lack of study consistence from students; but as a rule, the large majority of literature enacts both transformational and transactional leadership as the two most important leadership styles that play a direct or mediating role in regard to the higher education academic learning engagement, resulting as a rule in a mutual rewarding benefits atmosphere, based on trust and respect [105].

It considered together, the leaders and followers views, impressions and perceptions in regard to each other, along with each of the practiced leadership style power and influence over the students, lead towards a high-quality work environment that naturally, impacts not only the leadership effectiveness over learners, but the students learning engagement enhancement on the entire length of the academic framework. Given the previous arguments, the following main Hypothesis arises:

H2. Learning engagement has a positive effect on leadership

## Materials and methods

### Participants and procedure

By agreeing to the current study, a number of 2272 respondents, 18–35+ years old, freely agreed to take part and deliver a fully filled in questionnaire. The interval for the questionnaire entries covers 12/18/2022-05/19/2023. Participants were only required to take part on the

current research if they were active enrolled students within one of the legally accredited Universities in Romania, at the time. As form a gender point of view, the participants showed a distribution of 64% female and 36% male, currently undertaking a Bachelor, Masters, Doctoral and/or Postdoctoral fellowships, within the II[nd] semester of the 2022–2023 academic year.

## Methods setting and sample

By relying on the pragmatism and practice of previous research, the designated study relies on a convenience sampling, and considering the sampling method based on a voluntary response literature [106–109]; due to the technicalities of the research and its coverage, ease of access was considered with utmost importance. Google Forms platform was used as for gathering data, the questionnaire being presented to students covering a large variety of studies areas and specializations from Romania; as for reaching the respondents, numerous electronic communication platforms were used. The research considered and complied with the General Data Protection Regulation (EU Regulation 2016/679), since by filling in the required data, respondents were guaranteed for no personal data retention and/or requirement and provided with the answer's confidentiality and usage only for academic research purposes [110]. The participants, by freely agreeing to fill in the questionnaire, expressed their consent in regard to the previous information. No compensation was provided/nor implied by the researchers for the participants for filling in their input.

Respecting the requirements of the convenience sampling methodology, the sampling presented an easy way to identify and understand the defined research variables; as for defying the sampling bias, the current research methodology clearly states that respondents were only asked to provide a completely filled in questionnaire if they were subscribed to the active university enrollment requirement that was previously explained.

The research instrument design was intended to undertake and provide a deeper understanding on the link between higher education digitalization and students learning engagement under the actions and techniques specific of the academic leadership umbrella. The respondents were not provided with the receiving of any compensation or benefit from agreeing to taking part in the current research and providing a completely filled in questionnaire. Due to the usage of an exclusively online questionnaire distribution and filling out, data received covered a wide range of gender or age distribution and study enrollment hazard. The response rate was low, a number of 2272 accepted responses were gathered within a 6-months timeframe.

Romanian higher education academic specificities display a number of 45 civil higher education institutions and 34 private accredited higher education institutions. The list is being enriched with a number of 3 provisional authorized private higher education institutions, according to the same source [111].

The sample data reveals that actively enrolled students from a total number of 30 public and no private universities agreed to take action and fill up the questionnaire (therefore representatives from 66.6% of the public higher education institutions in Romania were involved).

## Measures

Since the questionnaire was distributed online via Google Forms, it's amplitude was reduced in scale for maximizing the respondents outcome and avoiding bias resulted from incomplete answers.

The research design provides four parts: after initially requesting simple demographic data and information in regard to the University they activate within, respondents were asked to provide answers in regard to three theoretical components: Higher Education Digitalization

through its five components (Internet and Digitalization Attitudes–IDA, Equipment and use of digital services for educational activities–DSEQ, Social media and online services used for educational activities–OSSM, Using Digital Services for Learning Purposes–LDS, Expectations for teaching and learning in higher education–LEXP); Learning Engagement was assessed through a number of three variables (Autonomy–AUT, Social Support–SS, Educational Activities Engagement EAE) and Leadership that according to its typicality, assumed a number of three forms (Transformational–TL and transactional through Contingent Reward–CR and Management by exception–Active–MBEA). It is important to stress the fact that the transactional leadership component only refers to the active management-by-exception since unified literature views do not consider management-by-exception passive to be a leadership situation; moreover, laissez-faire leadership views consider it as undistinctive and doubtful when compared to management-by-exception.

For the entire research instrument, a 7-point Likert Scale was used, thus replacing the traditional 5-point view; arguments supporting the current choice emphasize the increased accuracy, especially for online filling out environments, doe to the variety of provided choices. Therefore, for the current case, the 7-point Likert Scale choice was chosen for its suitability, thus surpassing the traditionality of the other similar restrictive choices. According to previous literature, a wider scale not only provides an increased number of options for the respondent, but also enhances the individual expressing possibilities, thus leading towards an increased results reality from appealing to their increased reasoning faculties.

Higher Education Digitalization consisting of a number of six variables was displayed with the help of a 7-point Likert Scale raging 1 to 7 (totally disagree/agree) and derived from the Austrian DigComp 2.2 AT Competence Model [16, 112]. The average internal consistency for the global variable was 0.85, and thus meeting values of 0.55 for DSEQ (5 items), 0.68 for IDA (6 items), 0.61 in the case of LDS (5 items), 0.71 for LEXP (6 items) and 0.47 for OSSM (6 items).

Learning Engagement followed the pattern off a 7-point Likert Scale ranging 1 to 7 (totally disagree/agree) and adapted from the previous 9 item Utrecht Engagement Scale. The average internal consistency was 0.92, with 0.81 for AUT (3 items), 0.82 for SS (3 items) and 0.91 in the case of EAE (8 items).

Both transformational and transactional leadership components derived from the MLQ (the 5X form), originally consisting of a 45 -item number, initially presented in 1995 and further developed [113]. The items display was adapted from a 5 to a 7-point Likert Scale, ranging 1 to 7 (totally disagree/agree). The internal consistency of the Leadership component averaged 0.87, with 0.86 for TL (5 items), 0.76 MBEA (3 items) and 0.79 in the case of CR (3 items).

## The analysis strategy

The strategic development of the current manuscript started with a general assessment of the three constructs and 11 subconstructs, by using SmartPLS (v. 4.0.0.) software. The analysis initiated with the intention of assessing the newly proposed research instrument in terms of internal consistency and reliability. Since the construct was successfully considered as valid, the analysis continued with a SEM analysis, a Confirmatory Tetrad Analysis and thee Importance Performance Map Analysis; after studying the FIMIX segment analysis, the specific indirect sizes were tested, and ended with testing the structural hypotheses.

According to the research objective, a full framework analysis of the Higher Education Digitalization–Students Learning Engagement–Leadership was provided and validated with a representativity of more that 65% of the state Universities in Romania.

Smart PLS software and analysis was carefully selected as for providing both authors and (re)viewers with a better understanding of the situational and regional technological advancements in regard to the studied variables [114, 115]. The current software is mainly recommended for cases including aa minimum of one formative construct, thus creating a partial least squares algorithm which provides two models: the outer model which is designed to provide data for the observable variables, yielded to the latent variables and the inner model that is designed to provide data in regard to a structural model that involved the proposed (observed) variables to another latent variables [116].

The current analysis considered at first as suitable the analysis of the outer model in terms of validity and reliability [117]; the inner model analysis mainly refers to path coefficient values analysis when in connection to other designated variables.

## Methods setting and sample

Emphasizing the practical and pragmatic a priori literature findings, the current research used a convenience sampling that has been undertaking a voluntary response [118]. Data gathering involved the online Google Forms Platform that generates a hyperlink used to further online distribution via various online (in)formal communication platforms, thus providing access and an easier reach for the future possible respondents. The research instrument design involved within its primary reach the General Data Protection regulation (GDPR), thus informing the respondents with the provided guarantees in regard to any requirements and/or usage of personal data; moreover, a strict confidentiality of the provided information and an exclusive academic usage was safeguarded.

The research carefully considered at first a pilot study, developed within 02–16.12.2022 timeframe, and after its validation, it was followed by the hypothesis's generation. As previously stated, the research is based on a convenience sampling method that implies a voluntary response. The text of the survey allowed the target audience with a sound understanding in regard to the identity and meaning of the variables; as previously mentioned, were considered to be valid only questionnaires filled in by actively enrolled students coming from public universities in Romania, without restrictions in regard to fields of study or degrees. Any other data was considered as not valid, and therefore excluded.

The research instrument design regards the amplitude of the students learning engagement phenomenon, highly engaged within a global quotidian exposure to digitalization information, communication and action means, under the umbrella of the traditional leadership guidance, which is subject to fading its actions, unless properly adapted. The three variable concept assesses therefore the connection between Learning Engagement (three variables), Higher Education Digitalization (five variables) and Leadership (three variables).

By agreeing to take part in the current research and fill out all the survey items, respondents consented to the lack of any type of fiduciary compensation. The survey was posted and distributed via online networks, within a five-month interval (December 2022-May 2023), gathering a number of 2272 valid answers. However, after a careful analysis, the response rate was considered to be low.

## Results

Presuming earlier results accepted by literature, a new research instrument assessment revises a number of steps out of which the first is the collinearity of the construct [119]. Therefore, the VIF values that characterize the construct collinearity severity values as compared to the specific thresholds of the SmartPLS software [120–122] meet the requirements of being lower

than 4 (<4.0) for the current case or 5 (<5.0) as literature suggests for some other cases, proving that the current database doesn't meet the conditions for collinearity issues.

The AVE (Average Variance Extracted) values specific to the convergent validity were measured by performing a consistent PLS SEM algorithm, factor, standardized. According to literature, the AVE thresholds should meet the condition of ranging above 0.5 (>0.5), previous literature recommendations [121, 123–125] suggesting for indicators with values lower than 0.5 to be dropped; as for the indicators with values within the range 0.4–0.7 [126], previous results indicate that they could be retained only by satisfying the condition for CR and AVE not to be affected by their presence [127].

According to the current results, a number of six items did not meet the criteria (IDA1 = 0.481; IDA3 = 0.101; IDA7 = 0.065; IDA 8 = 0.352; LEXP5 = 0.270; LEXP7 = -0.036). As for measuring an item convergence, the CR (Composite Reliability) along with Rho_c values must be used, by meeting the threshold of >0.7 in order for the variable (s) to be retained.

Construct reliability and Validity assessment must be considered by reporting values that meet satisfactory criteria for Cornbach's Alpha and Composite Reliability (CR) values, thus meeting the threshold of >0.6. As for the current initial model, both indicators ranged from 0.5 to 0.9, but after excluding the abovementioned indicators, values moved, thus ranging from 0.6 to 0.9, thus meeting the criteria for a reliable and valid construct.

Complementary to the previous traditional views, previous opinions in regard to Smart PLS software analysis [128–130] agreed over reporting Rho_A values (thus replacing the Cronbach's Alpha) when performing the reliability test; as for the current results, Rho_A values range from 0.6 to 0.9, thus confirming the reliability of the proposed database and proving its sufficient convergent validity (S1 Table)

By performing a PLS SEM analysis, data revealed the fact that the standardized coefficients values range from 0.1 to 0.6 and subscribing to the absolute declared value of <|1, providing data for the lack of multicollinearity issues for the current model.

As for the recommended steps for the assessment of a new model, the procedure requires the analysis of $R^2$ values [116]; as literature suggests, thresholds of 0.1, 0.3 and 0.6 are weak to substantial [131, 132], while according to different authors [133, 134], $R^2$ values peaking 0.0–0.2, 0.3–0.4, 0.5–0.6, 0.7–0.8, 0.9–1 describe ranges from weak to very strong. On another view only applied within behavioral sciences, the effect size "r" peaking 0.1, 0.3 and >0.5 ranges from small to large. Unanimously, the literature suggests that for particular constructs, the explained variance $R^2$ must meet values > = 1 in order to be considered adequate. Smart PLS software $R^2$ values are characterized by intervals peaking 0.25, 0.5, 0.75 and 1 respectively [121,135, 136], thus displaying a weak towards a substantial value for the explained variance. The current proposed construct displays $R^2$ values ranging from 0.99 to 1, thus substantial values. S2 Table

The imperfect nature of social sciences relates to the difficulty of accurately predicting human behavior, being difficult for a singular proposed model to be able to capture all the factors that accurately describe human behavior at a specific place and point in time [137]. R square values that subscribe to the 0.5–0.99 are considered as acceptable across social sciences research; the nature of the explanatory variables, as being statistically significant–as the model shows, and lack of multicollinearity, promotes the idea of a statistically significant explanatory variables within the given model.

The Model Fit criteria considers the SRMR (Standardised Root Mean Square Residual) Values for both models, estimated and saturated; the prudent range [138] must meet the threshold of <0.1, while the conservative view [139] provides the <0.08 value; as for the current case, for both estimated and saturated model, SRMR value meets the prudent criteria (= 0.1), thus presenting the proposed model with a good fit.

By performing a Confirmatory Tetrad Analysis (CTA-PLS), the model indicators were tested to see whether to be confirmed as reflective or formative. The current analysis cannot be performed for latent variables containing less than four associated indicators; the p-value within the current case has a threshold of 80% of the p-values of >0.05 to be best reflective, while for values <0.05, the variable is considered to be best formative. CTA-PLS analysis was performed with two-tailed parallel processing, 10 000 subsamples; the results show DSEQ, EAE, IDA, LDS, LEXPM and OSSM to be best formative, while TL, Leadership, Leadership engagement, Higher Education Digitalization were considered to be best reflective.

The inner model heterogeneity was further tested, by using the FIMIX-PLS analysis which is considered to be the best suited when assessing the possibility of hidden segments [140]. The latent class analysis is considered to be best suited in cases when a priori prevention methods could not be applied for external sources [140–142]. The assumption for the current analysis consists of an 80% power level *with a 0.15 effect size); given the dimension of the current database, the analysis should be performed on a number of approximate 40 extracted groups [143]. By considering the segment sizes, by performing the analysis for 6 segments, data shows a downgrade trendline which is assumed to continue. The results show that 47% of the data is in segment 1, the rest spreading across the other segments (S3 Table)

As for determining the number of the retained segments and their separation, for the current database, a number of criteria are to be considered: AIC (Akaike's Criterion) modified with factor 3, EN (the Normed Entropy statistics), Consistent AIC (CAIA) were used [144–146]. The EN takes values 0 to 1 and offers information regarding the segment's reliability; a higher value reveals a higher partition quality [147]. Another literature view suggests that if EN takes values >0.5, the data will be allowed to be cut from any other predetermines segments [148].

According to initial results, data is not minimized to a single segment (S4 Table)), but EN does not take values above 0.5 thus excluding the possibility of having multiple segments. One final step suggests considering the Discrete Segments Assignment as a final indicator, whose values do not load over the threshold of 0.2; combined data suggests that no strong correlations have been observed with the segmenting variable, resulting that the current database does not show multiple segments. Data in regard to the results robustness are considered to be therefore, satisfactory [149].

The Q2 values revealed by the PLS predict analysis [124] has the role of comparing the path model prediction errors to the simple mean; as recommended by previous analysis results [150], the evaluation of the PLS SEM model requires the current model predictive power. Results show that for the Q2 >0 for all the values, the model has been correctly specified and has a good predictive power for the inner model (S5 Table).

By performing the Process emulator with quadratic nonlinear effects analysis, two of the variables (CR and TL) were used as control variables, therefore adding regression lines to the endogenous variables (Higher Education Digitalization and Learning Engagement). The model specifics as formative or reflective variables are not substantiated within the current analysis. As for the current model, it was considered that MBE_A is a moderator for all the direct effects of Higher Education Digitalization and all its variables. A quadratic effect was also added to the Higher Education Digitalization—Learning Engagement relation (S6 Table)

Results show no conditional direct effects as significant; given that all the p values for the conditional direct effects are significant, results show for example that at mean, the moderating effect of MBE, the mediator effect from DSE to HED is statistically significant.

The specific indirect effect sizes consider the v2 thresholds for small, medium and considerate for 0.1, 0.04 and 0.09 thresholds [151]. Results considering values above p>0.05 threshold display too much error, thus determining rejection; for the values of p<0.05, the result is considered as significant, thus supporting the existence of an observed effect. For situations where

v<0.01, the context (e.g. the small size of the sample, the underdeveloped research subject) must be considered, only for the cases when the p value is statistically significant and the effect (v) displays values greater than 0; such a context must be interpreted as a signal for the possibility of future research and development in regard to the subject area (S7 Table). The lower thresholds are mainly suitable for sociological or psychological fields of study, as the literature suggests [152].

An (extremely small) effect size can be demystifying the signal of a an existing/initiating process; in this case, the resulting information could be interpreted under the format of a significant contribution for developing and supporting further one or more predictions of a future event, specific for the studied area [136, 153]. Such a phenomenon could be interpreted as the process of territorial systemic digitization of the higher education, that could act as a predictor for the future degree of students' engagement, while actively enrolled within the processes specific to the higher educational curricula [154, 155].

Results only show a single small statistically significant effect for EAE -> LEARNING ENGAGEMENT -> LEADERSHIP.

Considering the outer model to be valid, the structural hypotheses will further be tested. Since not all the factors are reflective, a simple PLS SEM and Bootstrap procedure were performed, where thee $R^2$ and the $F^2$ values were considered (Fig 1).

By considering the absence of a direct effect between the Higher Education Digitalization and Learning Engagement, a number of 14 structural hypotheses was tested (S8 Table)

In the light of previous research, it was assumed the dimensions of Learning Engagement is directly affected by the Higher Education Digitalization. Results show that all the leadership variables (TL, CR, MBEA) lose their prediction accuracy for the primary Leadership construct. All the three hypotheses have a small effect, thus losing their significance (S9 Table).

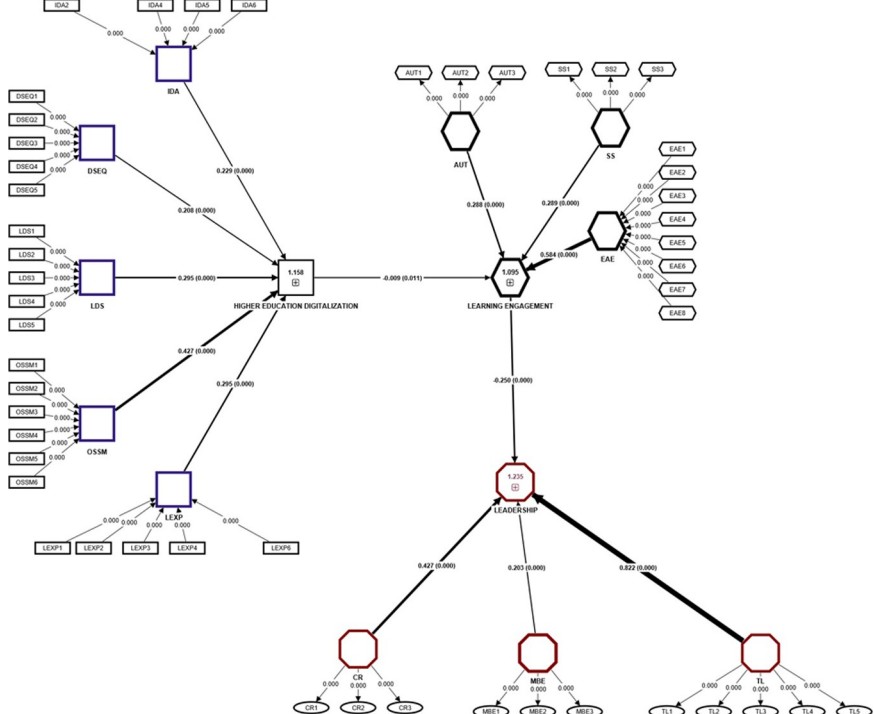

**Fig 1. Consistent PLS Bootsrapp for testing structural hypotheses with highlight using relative values.**

In the light of previous results, an Importance Performance Map Analysis was performed. On the X Axis results display their predictive power, while on the Y axis, the relative influence that the predictor has on the outcome target variable may be observed.The analysis facilitates the identification of the variables which are good predictors and the analysis of how well they perform; the analysis targeted at first Learning Engagement, followed by Higher Education Digitalization and Leadership.

EAE is the best predictor for Learning Engagement and performs the best; out of the entire variable, with utmost importance is EAE2 indicator (the higher in importance), followed by EAE 7 which is the highest in performance, followed by EAE4, EAE3 (Fig 2).

SS and AUT are far differentiated by their previous competitor, with a slight increase in performance for SS. For the two variables, indicators as SS2, SS3, SS1 and AUT 3, AUT2 and AUT1 are the most important.

The best predictor in importance for Higher Education Digitalization is OSSM, followed by LEXP (which leads in performance) and LSD, while the least important predictors are DSEQ and IDA.

OSSM4 is the most important predictor in terms of importance, followed by IDA2 and LDS1 which also lead in performance (Fig 3).

The backwards scale is constituted by indicators as DSEQ2, DSEQ1, OSSM3, LEXP1, LEXP6, IDA5 and LEXP2, with very close variations on both scales.

As for the Leadership variable, the most important predictors are TL in both importance and performance, followed by CR and MBE (Fig 4).

TL5, TL4, and CR1 cover the same are on both dimensions, closely followed by TL3, CR3, TL2, CR2, almost equally important and performing similar.

The analysis also includes TL1, followed by MBE1, MBE2 and MBE3 in reverse order of importance. As for the rest of the items, they perform very low in importance, with one exception: LDS1, LEX4, SS1 are very good predictors in performance for the Leadership variable.

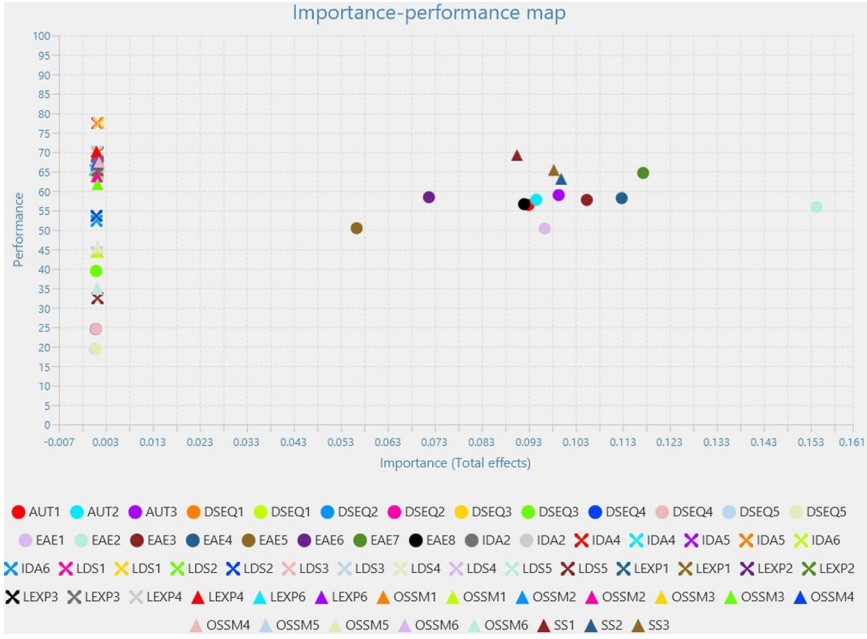

**Fig 2. Learning engagement importance performance map analysis.**

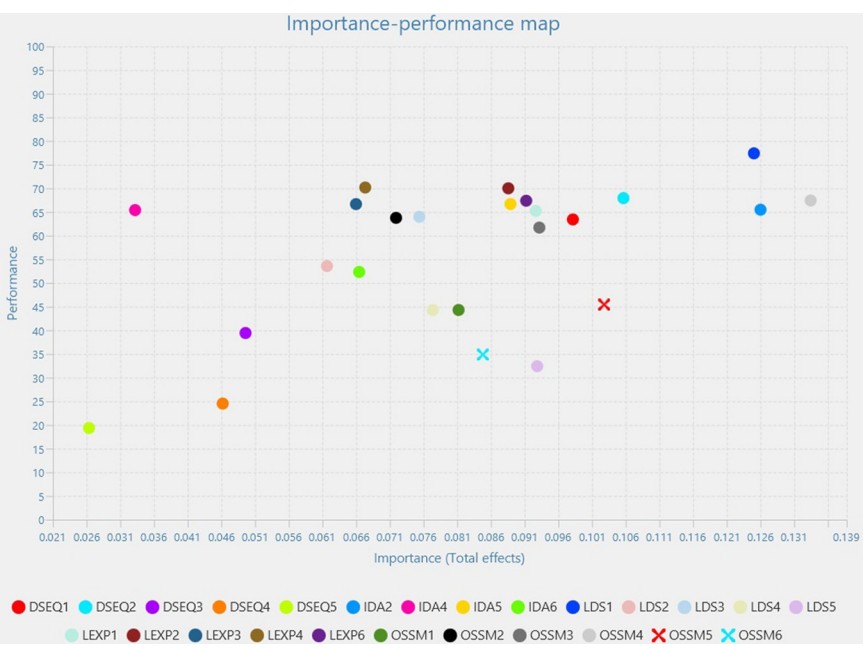

**Fig 3. Higher education digitalization importance performance map analysis.**

The Importance-Performance map analysis has revealed the fact that EAE is the best predictor for Learning Engagement and performs the best, while the most important predictor in importance for Higher Education Digitalization is OSSM. Moreover, the Leadership variable is best represented first, is expected, by TL, followed by the transactional leadership components–CR and MBE.

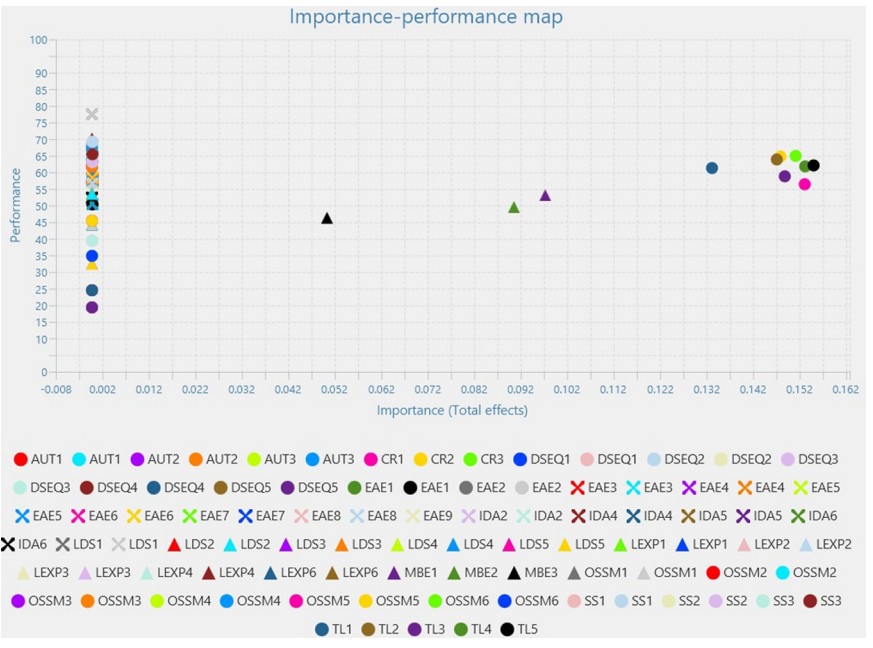

**Fig 4. Leadership importance performance map analysis.**

## Hypothesis testing

The first hypothesis states that Higher Education Digitalization has a negative effect over the students learning engagement, under all their components. As testing the model and analyzing the path coefficients, it was analyzed whether the current hypothesis is entirely valid; according to the current model factor loadings, the HED (Higher Education Digitalization) in relation to the LE (Learning Engagement) has the highest value for OSSM (0.337), followed by LEXP (0.281) and LDS (0.278); the lowest value has been assimilated to DSEQ (0.209), but without affecting the validity of the H1, which has been entirely supported by data.

The current study assumes the existence of a global variable Leadership, comprising two general leadership styles (transformational and transactional) and three variables–TL, CR, MBE. Further, the authors tested whether there is a connection between Learning Engagement and the global Leadership variable. According to the factor loadings, Learning Engagement has a positive effect on Leadership, therefore supporting the second general hypothesis of the current study (H2).

In between, all the latent variables were tested in regard to the strength of connection to their own components; results have shown that all Higher Education Digitalization variables (IDA, FDSEQ, LDS, OSSM and LEXP), along with Learning Engagement variables (AUT, SS and EAE) and lastly, Leadership components (CR, MBE and TL) have proved to have strong effects, thus confirming all the proposed secondary hypotheses (H3-H14).

The further section referring to a general discussion of the results in the light of previous research cannot follow without emphasizing the fact that by considering the literature theories [156], all thee six elements of the proposed Higher Education Digitalization variable have been confirmed, after testing it for CR and variance. Therefore, it proved to be a valuable assessment version for the higher education organizational environment, providing valuable insights for further digital developments and improvements. Developed as a supporting study for a previous initiative [157] the higher education digitalization assessment gained and stressed the importance of managerial specific practices, organizational traditions, investment strategies, all leading towards achieving a higher individual and institutional performance. By assessing not only a singular organizational higher institutional framework, but by referring to a large percentage of the country institutional representatives, the proposed study could gain interest for both singular individual institutional strategic views, but also on a wide conglomerate (ministerial) setting.

## Discussion

According to the twofold aim of the current study, the manuscript proposal was initiated with the validation of a new research instruments that regards the actively enrolled higher education student's engagement within their derivative requirements, duties and responsibilities. Further, as for the institutional personnel perspective, two types of leadership (as transactional and transformational) were proposed, as for explaining the features and behaviors of academic staff, thus acquiring a leadership perspective. Additionally, the Higher Education Digitalization was considered to have a moderating role for the relationship involving transactional (as CR and MBEA) and/or transformational (as TL) leadership and the Higher Education Digitalization. The Romania higher education institutional context is characterized, as the results show, to be a negative influence for learning engagement (see H1 and the derivative hypotheses), while students learning engagement is positively connected with academic (staff) leadership (see H2 and the derivative hypotheses). Ultimately, according to the PLS Predict values, MBE_A is a moderator for all the direct effects of Higher Education Digitalization and all its

variables, thus emphasizing the role of academics for not focusing on error, but empowering students to deal with such errors and reach and overcome preset standards [158].

As for the importance of the current study, emphasis is put on the singular approach of the Romania Higher Education environment (thus including a significand amount of universities) from an active enrolled students perspective in regard to digitalization processes, presumed to act as a facilitator for students engagement within daily required tasks and academic leadership. As for a better understanding of the current perspective, further, results and findings will be discussed at length.

## Learning engagement and leadership

As the current research evidence, most of the literature that assesses the leadership (as transformational and other assimilated forms) and learning engagement within an educational framework, mainly referred to the Bass and Avolio's transformational leadership, literature providing little examination in regard to other leadership styles. Moreover, transactional leadership approaches mainly focus on the academic leaders approaches of the students' decisions to study, based on the premise that circumstantial personal factors specific to students lead towards systematic approaches in regard to learning [159–161]. Further investigation revealed the fact that based on their characteristics, other leadership styles (such as transactional or laissez faire) could benefit from association to learning engagement, in both business and higher education organizational environments, thus leading to an active broadening of empirical research in this regard [162].

As for the current results, the specifics of the higher education system reversed the sphere of influence; from the instances where leadership was the main variable in the relation of the students with the academic requirements, the novelty of the current research is that higher education learning engagement fairly influences the practice of leadership styles within the given institutional framework. Learning engagement consists of the students' own attitude towards learning, self-guidance, reason for which EAE reveals the higher loadings within the model; this fact could be explained through the fact that personal willingness and self-motivation weight more for the students than leaders (as academic staff) strategies and attitudes; therefore, students which feel strong, bursting with energy, feel happy when studying and get carried away, are hard to be subject to any leadership styles that could improve their state of mind; in reverse, academic staff that in presented within study hours with such attitudes, find themselves as changing leadership attitudes and practices; the assessment and implementation of a new leadership style, adequately applied to self-motivated learners, becomes a new issue for academics, gaining in importance, in the context of the evolutionary higher education digitalization. Moreover, learning engagement presents other two components that are less assessed across literature: social support and autonomy. Higher education digitalization frameworks allow an enhanced communication among learning communities, therefore the social support provided by academics gains a new feature: speed—of reply, of feedback, of solving arising community issues. Autonomy on the other side, allows flexibility, control and decision making in regard to the students; learning process, leading towards an increased engagement in regard to the daily tasks. So often, from a myriad of variables that could influence the academic leadership, students learning engagement gains in importance, transforming from a secondary to a primary variable to which higher education systems should pay attention, in the context of a rising digital environment. Further developments could focus on the leadership styles which are more susceptible to be changed by the digitalized higher education learning environment; new strategies should be adopted, if leadership is to continue to be prevailing over the organization and functioning of organizational systems. Based on these explanations

from various studies and literature cited above, the current hypothesis in the study is that in the continuous process of higher education digitalizing system, students learning engagement has a significant effect on academic leadership.

## Learning engagement and higher education digitalization

Learning engagement resides within a self-motivation process that provides access to the possibility to learn, study or practice by using academic resources, leading towards enhanced personal and institutional outcomes; these come under the form of learning engagement stimulation, transforming the higher education environment into a healthier and more productive holder [163]. The current research considered at first the idea that a digitally satisfied learning student would be an asset for the further endeavors of the academic environment [164]; previous research, ante and post Covid-19 pandemic considered learning engagement and productivity to increase, as the satisfaction increases. Despite these general views, the current results show that the level and ways of digitalizing the academic institutional framework lead towards a negative effect on the students' learning engagement. Specifically, OSSM, LEXP and LDS (as Higher Education Digitalization components) have been proven to have the highest effects on students learning engagement, out of which EAE, followed by AUT have been more engaged. SS, the third component of Learning Engagement, has previously been proven by numerous studies to bear the role of a mediator when about leadership and engagement [165], therefore the supposition is that within the current research, it's function has not been subject for significant change. The reasons behind the newly developed higher education systems' specific scene derive from the direct effects of the recent Covid-19 pandemic, when due to civil travel and movement restrictions, all the higher education institutions were forced to provide fully adapted digitally instruments for the development of the academic year. Given the fact that data was collected at approximately two years after the endpoint of the special circumstances, the general view over how the system managed to work shows deeply perceived gaps, reminiscences of the period. Not being able to adapt efficiently, higher education digitalization process provided permanent damages within the students' assessments in this regard. Follow-up and assessment activities following the pandemic period appear not to be sufficient. For these reasons and the explanations provided by the literature cited above, the second hypothesis of the study explaining that in the light of leadership strategies and practice, the digitalization of the higher education system has a negative effect on the students' learning engagement.

## Theoretical implications

The current manuscript's contribution to literature initiates with the proposal and validation of a higher education digitalization assessment instrument comprising six initial variables, out of which five could be acknowledged. According to literature [166], the role of contextual studies is to bring research outputs closer to the needs and realities of institutions representatives, members and peers; the current endeavor has been emphasized by the fact that the respondents replies involved within the current study were only selected if being actively enrolled within a higher education program, thus reducing the risk from recall bias. Given the current technicalities, the provided results referring to the digitalization degree of higher educational teaching and learning environment is considered to provide accurate and reliable results, compared to analytical situations that reflect recall situations of a specific moment in time.

As previous observations suggest, the current research is one of the few of its kind to provide clear data in regard to the digitalization degree of higher education institutional environment, in the context of two leadership styles practice (as transformational and transactional)

and three variable students learning engagement component. Research findings suggest transformational leadership and expectancy to learning engagement (EAE) are the higher predictors for higher education digitalization. Previous results suggest transactional leadership to be a key predictor to higher institutions digital transformation, along with the SS component of learning engagement. Moreover, literature suggests that given a common institutional context, work engagement could be easily stimulated by CR [167], a contrary opinion [168] proved that transactional leadership does not have the practical potential to alter work engagement. Given the current theoretical standpoint in regard to transformational leadership theories and practice, its power prevails over the transactional form, situations revealing its role of augmentation for the latter leadership style considered [169]. Aside from particular situations, as literature suggests transformational and transactional leadership behaviors can be recognized with different practice attitudes and outcomes [170]; the current results support the given hypothesis, the traditional transactional and transactional leadership roles and characteristics prevailing over the results. Aside form digital transformation and digitalization endeavors within the higher education environment, it is worth to mention the arising needs of the accelerated future developments that include the technological research and communication defiance of intelligent research technologies, that might create disruptions in the practice of academia leadership styles and more carefully, within the behavior and thinking patterns of students, resulting into attitudes of disengagement rather that engagement.

The two-style leadership assessment of a national higher education framework might prove to be useful under different levels of management and decisional prospects, while the relation considering system digitalization and students engagement could make room for further necessary investments. Further, the practical applicability of the current results could result into increased institutional and individual increased academic performance, for both leaders and followers, thus promoting students' developments and performance achieving if and when engaging into a state-owned higher education institution.

## Practical implications

As an initial practical standpoint, the current results emphasize the importance of applying an accurate digitalization assessment instrument, that would lead to further informed investments decisions, by measuring trajectory and performance.

Further, by assessing the linkage between the practical use of digital tools for enhancing the students learning engagement, provides vital information for future singular and the entire system of higher education institutions for further initiatives that would lead to increased student retention and academic performance, but also increased subscription rates and also insertion on the labor market.

The current developments could not prevail without addressing the leadership issue in regard to styles practice and achieved performance, as perceived by followers; when assessing the given questionnaire, the respondents were asked to regard leadership only in relation to their teachers and academic staff, thus the current results provide an accurate picture of the teaching-learning results in the context of the entire higher education system in Romania. As expected, transformational leadership was recognized with utmost importance when assessing its effects on students' engagement; but unexpectedly, the relationship higher education digitalization–students learning engagement received negative hidden meanings.

Previous research confirms increased leadership performance under institutional training programs [171, 172] therefore ministerial initiatives could be issued for nurturing and preserving transformational leadership initiatives for every institutional representative of the system. Results could favor the creation of an educational system institutional map that could correlate

leadership training investments to practical students results, thus favoring their insertion into the labor market and exceeding their chances for achieving performance across industries.

The subject of adapting the traditional leadership tools to a highly digitalized teaching and learning environment arise across literature, since an adapted leadership solution may break the performance barrier and raise the abilities to foster solutions for teaching added value and innovativeness. The context of a socially awaken and digital student forces leaders to innovate and comply to novel teaching solutions that would lead towards a prosperous systematic organizational change, where public universities as host organizations, are to be engaged as continuously and perfectly adapted institutional cradles that foster a digitally adapted academic performance and high levels of individual students' engagement.

## Study limitations

By setting aside the strengths of the current results that can be relied upon, the considered limitations bring into light at first literature views in regard to the use of qualitative assessment tools that present an increased risk for bias, usually reflected within the variance of the statistical results [173]. A contrary opinion [174] claims that common assessment methods cannot be subject to bias, since the currently proposed research construct only reflect moderate relationships. Additionally, the anonymity of the respondents reduced the risk of any potential bias.

Since only using two transactional leadership components (and excluding MBE passive from the research), a second limitation could arise. As literature suggests that not all the leadership components always relate to the cultural and organizational environment they are studied within, the current research could be counted as providing an inverted academic environment that instead, analyses key aspects of transactional and transformational leadership practice. Literature suggests that MBE passive was mainly found present within organizational environments characterized by monitoring mistakes with large statistical effects; given the lack of data availability for the respondents for the side of institutional representatives, for the benefit of the current study, the MBE passive leadership style was not considered as suitable for assessment. As for completing the national institutional panorama, further enhancements would not only consider MBE passive for analysis, but also the private higher education institutions for analysis, thus making room for all the leadership styles as defined by Bass and Avolio [10] to unveil [175–177].

As for the context characterization, further research could include a specific representative number of students coming from each public and/or private institution or, more in depth, coming from similar specializations and study fields.

Given the number of questionnaire replies, the reliability and validity, along with the stability of the construct design has been proven; naturally, such results could provide a operational working platform for any ministerial and/or institutional initiatives for further developments and analysis. Therefore, a question might arise in regard to the specifics of the digitalization advancements for each institutional peer involved, thus leading to different levels and nature of students' engagements and leadership practice. as prior literature suggests [178] the answer could only be revealed by analyzing the managerial and leadership practices of administrative peers, since leaders influence greatly varies according to the institutional backup and resources [179].

## Conclusions

To our knowledge, the current manuscript contributions to literature is one of the first attempts to validate an assessment research instrument that regards de digitalization degree on a national higher education institutional framework. Additionally, the results emphasize the importance and the novelty of the digitalization assessment in correlation to the practice of

two leadership styles, in the context of in(de)creasing individual students learning engagement. The analysis provided information according to which the digitalization of higher education institutions (as opposed to traditional teaching and learning tools and practices) is a negative driver for the engagement of actively enrolled learners' engagement is performing their study tasks and requirements. Furthermore, learning engagement and leadership play a constructive role within the higher education institutional cradle, particularly the EAE component which has a small effect of the general leadership variable. Additionally, by performing the PLS predict Analysis, results considered that MBE_A is a moderator for all the direct effects of Higher Education Digitalization and all its variables. To conclude, the current research stands for a functional and sustainable research platform that provides incentive for present or future institutional or ministerial incentives in regard to digitalization necessary endeavors and leadership training and development, as for improving the teaching-learning practices that would lead to higher levels of individual students learning engagement. Such an initiative provides a standing framework for institutional insight for investments of social endeavors, enhancing altogether the future of the workforce members for an easier insertion on the labor market and increased and advanced individual and group performance.

## Supporting information

**S1 Table. Construct reliability and validity (final values).** Final values for the construct reliability and validity.
(TIF)

**S2 Table. Construct R^2 values.** Construct R^2 values.
(TIF)

**S3 Table. FIMIX segments sizes.** Reference to the explanatory values of the six considered segment sizes.
(TIF)

**S4 Table. FIMIX results.** Explaining the considered values for the FIMIX results.
(TIF)

**S5 Table. PLS predict values.** Explaining the values for the PLS Predict analysis.
(TIF)

**S6 Table. Conditional direct effects.** Explaining the conditional direct effects.
(TIF)

**S7 Table. Specific indirect effects.** Explaining the specific indirect effects.
(TIF)

**S8 Table. Testing structural hypotheses (part 1).** Explaining the first results after testing structural hypotheses.
(TIF)

**S9 Table. Testing structural hypotheses (part 2).** Explaining the results for testing structural hypotheses (part 2).
(TIF)

**S1 Database.**
(XLSX)

## Author Contributions

**Conceptualization:** Romeo Asiminei.

**Data curation:** Ioana Gutu.

**Formal analysis:** Ioana Gutu.

**Investigation:** Camelia Nicoleta Medeleanu.

**Methodology:** Ioana Gutu.

**Project administration:** Romeo Asiminei.

**Resources:** Ioana Gutu, Camelia Nicoleta Medeleanu.

**Software:** Ioana Gutu.

**Validation:** Ioana Gutu, Camelia Nicoleta Medeleanu.

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
