## [Decision Letter · Decision Letter 0]

8 Apr 2024

PONE-D-24-08799ANALYSIS OVER THE CONTEXT OF THE HIGHER EDUCATION DIGITALIZATION WITHIN THE LIMITS OF STUDENTS LEARNING ENGAGEMENT AND ACADEMIC LEADERSHIP BY USING PLS SEMPLOS ONE

Dear Dr. Gutu,

Thank you for submitting your manuscript to PLOS ONE. After careful consideration, we feel that it has merit but does not fully meet PLOS ONE’s publication criteria as it currently stands. Therefore, we invite you to submit a revised version of the manuscript that addresses the points raised during the review process.

The author needs to proofread the whole manuscript.

We look forward to receiving your revised manuscript.

Kind regards,

Chunyu Zhang

Academic Editor

PLOS ONE

Journal Requirements:

Reviewers' comments:

Reviewer's Responses to Questions

**Comments to the Author**

1. Is the manuscript technically sound, and do the data support the conclusions?

Reviewer #1: Yes

Reviewer #2: Yes

2. Has the statistical analysis been performed appropriately and rigorously? 

Reviewer #1: Yes

Reviewer #2: Yes

3. Have the authors made all data underlying the findings in their manuscript fully available?

Reviewer #1: Yes

Reviewer #2: Yes

4. Is the manuscript presented in an intelligible fashion and written in standard English?

Reviewer #1: Yes

Reviewer #2: Yes

5. Review Comments to the Author

Reviewer #1: The manuscript is well-organized, and the methodology and results are clear. Hence, I recommend the publication; after that, some minor suggestions will be addressed.

Comments

The definition of engagement and its types should be discussed in the Foreword section. This is given late in the article.

Was the informed consent obtained from the participants?

Reviewer #2: Kindly change in abstract

Most of the research in regard to improving learners’ digital skills that would ultimately result

into increasing their active engagement was performed over primary and secondary students, thus leaving the

higher education members aside.

The suggested change is as Under

The majority of the research that was conducted on the topic of enhancing the digital skills of learners, which would ultimately lead to an increase in their active engagement, was conducted on students in primary and secondary education, leaving members of higher education outside of the scope of the study.

Foreword should vie replaced by Introduction

H1. H1 HIGH EDUC DIGI HAS A NEGATIVE EFFECT ON LEARNING ENG

Remove ine HI It has appeared twice

Results display that R^2 values are very high. What could be the reason for this.

EAE -> LEARNING ENGAGEMENT -> LEADERSHIP Displays small effect. Kindly justify.

Support through literature

What could be implications of It

Kindly adjust references as per the journal style.

Instead of putting p=0

put them as p < .01.

The paper is good. These changes may be incorporated in the manuscript.

6. PLOS authors have the option to publish the peer review history of their article (what does this mean?). If published, this will include your full peer review and any attached files.

Reviewer #1: No

Reviewer #2: No

---

## [Author Response · Author response to Decision Letter 0]

10 Apr 2024

Reviewer #1:

Dear Reviewer,

Thank you for your kind support in regard to publishing our Manuscript. 

As a response to your valuable insights, we introduced engagement and it’s types early within the foreword. Thank you for your suggestion.

As for the informed consent, the entire process and Ethical agreement approvals were already provided within the early submission stages. We included the necessary information within the Methodology section. Thank you for your suggestion.

Thank you very much for your support and valuable insights as for improving the quality of our manuscript.

Best regards,

Reviewer #2

Dear Reviewer,

We provided the suggested change within the Abstract. Thank you for your suggestion.

We replaced Foreword with Introduction.

We removed the duplicate of H1. 

We explained with literature support the possible reasons for the high R^2 values.

We provided literature support for the small effect EAE -> LEARNING ENGAGEMENT -> LEADERSHIP, and discussed possible implications. 

We adjusted the references as per Journal style. 

We revised the entire methodology, including tables and figures.

Thank you very much for your support and valuable insights as for improving the quality of our manuscript.

Best regards,

---

## [Editor Report · Decision Letter 1]

15 Apr 2024

PONE-D-24-08799R1ANALYSIS OVER THE CONTEXT OF THE HIGHER EDUCATION DIGITALIZATION WITHIN THE LIMITS OF STUDENTS LEARNING ENGAGEMENT AND ACADEMIC LEADERSHIP BY USING PLS SEMPLOS ONE

Dear Dr. Gutu,

Thank you for submitting your manuscript to PLOS ONE. After careful consideration, we feel that it has merit but does not fully meet PLOS ONE’s publication criteria as it currently stands. Therefore, we invite you to submit a revised version of the manuscript that addresses the points raised during the review process.

The clean version is inconsistent with the tracked version. Please resubmit it.

We look forward to receiving your revised manuscript.

Kind regards,

Chunyu Zhang

Academic Editor

PLOS ONE
---

## [Author Response · Author response to Decision Letter 1]

25 Apr 2024

Dear Dr. Chunyu Zhang,

At first, I am expressing my most sincere gratitute for handling this manuscript. 

Altogether, I am thanking you for drawing attention on the requirements in regard to the previous submission.

I removed the initial manuscript, and added once again Only the Track changes and simple Manuscript versions. i hope that I understood correctly the requirement. 

Thank you once again. 

Best regards,

Ioana Gutu

---

## [Decision Letter · Decision Letter 2]

11 Jun 2024

ANALYSIS OVER THE CONTEXT OF THE HIGHER EDUCATION DIGITALIZATION WITHIN THE LIMITS OF STUDENTS LEARNING ENGAGEMENT AND ACADEMIC LEADERSHIP BY USING PLS SEM

PONE-D-24-08799R2

Dear Dr. Gutu,

We’re pleased to inform you that your manuscript has been judged scientifically suitable for publication and will be formally accepted for publication once it meets all outstanding technical requirements.

Kind regards,

Chunyu Zhang

Academic Editor

PLOS ONE

Additional Editor Comments (optional):

I suggest a conditional acceptance. Authors need to ensure consistency between title, purpose, research question, and H1.

Reviewers' comments:

Reviewer's Responses to Questions

**Comments to the Author**

1. If the authors have adequately addressed your comments raised in a previous round of review and you feel that this manuscript is now acceptable for publication, you may indicate that here to bypass the “Comments to the Author” section, enter your conflict of interest statement in the “Confidential to Editor” section, and submit your "Accept" recommendation.

Reviewer #3: All comments have been addressed

2. Is the manuscript technically sound, and do the data support the conclusions?

Reviewer #3: Yes

3. Has the statistical analysis been performed appropriately and rigorously? 

Reviewer #3: Yes

4. Have the authors made all data underlying the findings in their manuscript fully available?

Reviewer #3: Yes

5. Is the manuscript presented in an intelligible fashion and written in standard English?

Reviewer #3: Yes

6. Review Comments to the Author

Reviewer #3: After the two evaluation rounds, the work is brought to a reasonable standard from the scientific perspective. The authors adequately implemented all the recommendations in the previous reviews. The Introduction complies with the methodological expectations, and the 111 works consulted and referenced in the "Materials and Methods" section support research hypothesis H1.

However, I notice a slight inconsistency between the title, purpose, research question, and H1.

While maintaining H1, I propose a potential adaptation for the other elements. This suggested format could potentially enhance the clarity and coherence of your manuscript:

Title: "The Impact of Higher Education Digitalization on Student Engagement: The Role of Academic Leadership"

Aim: "This manuscript aims to examine how the levels of higher education digitalization affect student engagement in the learning process under the influence of academic leadership styles."

Research Question (RQ): "How do the levels of higher education digitalization affect student engagement in the learning process under the influence of academic leadership styles?"

Hypothesis H1: "Higher education digitalization has a negative effect on learning engagement."

Another recommendation is related to the elimination of the Introduction of the results reached by the study and an insistence on the purpose of the study.

The statistical part, discussions, and conclusions are relevant for H1.

Otherwise, I agree with the form in which the work is presented, and the paper can be accepted for publication of this paper after these minor updates.

7. PLOS authors have the option to publish the peer review history of their article (what does this mean?). If published, this will include your full peer review and any attached files.

Reviewer #3: No

---

## [Editor Report · Acceptance letter]

28 Jun 2024

PONE-D-24-08799R2 

PLOS ONE

Dear Dr. Gutu, 

I'm pleased to inform you that your manuscript has been deemed suitable for publication in PLOS ONE. Congratulations! Your manuscript is now being handed over to our production team.

Kind regards, 

on behalf of

Dr. Chunyu Zhang 

Academic Editor

PLOS ONE